# vHDvDB 2.0: Database and Group Comparison Server for Hepatitis Delta Virus

**DOI:** 10.3390/v16081254

**Published:** 2024-08-05

**Authors:** Chi-Ching Lee, Yiu Chung Lau, You-Kai Liang, Yun-Hsuan Hsian, Chun-Hsiang Lin, Hsin-Ying Wu, Deborah Jing Yi Tan, Yuan-Ming Yeh, Mei Chao

**Affiliations:** 1Department of Computer Science and Information Engineering, Chang Gung University, Taoyuan 33302, Taiwan; 2Molecular Medicine Research Center, Chang Gung University, Taoyuan 333, Taiwan; 3Genomic Medicine Core Laboratory, Chang Gung Memorial Hospital, Taoyuan 33305, Taiwan; 4Department of Microbiology and Immunology and Division of Microbiology, Graduate Institute of Biomedical Sciences, School of Medicine, Chang Gung University, Taoyuan 333, Taiwan; 5Liver Research Center, Department of Hepato-Gastroenterology, Chang Gung Memorial Hospital, Taoyuan 333, Taiwan

**Keywords:** web server, database, hepatitis delta virus, genotyping

## Abstract

The hepatitis delta virus (HDV) is a unique pathogen with significant global health implications, affecting individuals who are coinfected with the hepatitis B virus (HBV). HDV infection has profound clinical consequences, manifesting either as coinfection with HBV, resulting in acute hepatitis and potential liver failure, or as superinfection in chronic HBV cases, substantially increasing the risk of cirrhosis and hepatocellular carcinoma. Given the complex dynamics of HDV infection and the urgent need for advanced research tools, this article introduces vHDvDB 2.0, a comprehensive HDV full-length sequence database. This innovative platform integrates data preprocessing, secondary structure prediction, and epidemiological research tools. The primary goal of vHDvDB 2.0 is to consolidate HDV sequence data into a user-friendly repository, thereby facilitating access for researchers and enhancing the broader scientific understanding of HDV. The significance of this database lies in its potential to streamline HDV research by providing a centralized resource for analyzing viral sequences and exploring genotype-specific characteristics. It will also enable more in-depth research within the HDV sequence domains.

## 1. Introduction

The discovery of the hepatitis delta virus (HDV) can be traced back to the mid-1970s, when the delta antigen (HDAg) was first described in carriers of the hepatitis B virus (HBV) in Italy [1]. Transmission experiments in chimpanzees demonstrated that this new antigen was a defective virus, requiring HBV for infection [2]. It is named HDV and is recently classified in the family Kolmioviridae [3].

The HDV virion is approximately 35-36 nanometers in size, with an RNA genome length ranging from 1672 to 1697 nucleotides, making it one of the smallest viruses in human virology [4]. The virus is enveloped by the hepatitis B surface antigen, which it requires for entering hepatocytes, intrahepatic spreading, and transmission among hosts. The HDV genome is of negative polarity and is complexed with the delta antigen (HDAg) [5]. The RNA genome and its complement, the antigenome, are circular in conformation [6], and both harbor a highly conserved pseudo-knotted structured ribozyme [7]. HDV RNA can form an unbranched, rod-like structure, with around 70% of its nucleotides creating base pairs. [5,8].

HDAg is the only virus-encoded protein present within both virions and infected cells. The HDAg exists in two forms. In the early stages of replication, the small form (S-HDAg) is produced and is crucial for viral replication. [9]. During replication, RNA editing of the antigenomic HDV RNA induces a mutation that changes the amber stop codon of S-HDAg into a tryptophan codon (W), facilitated by the host RNA adenosine deaminase, ADAR1 [10]. This editing process leads to the creation of a larger HDAg (L-HDAg) that has an extra 19-20 amino acids at the C-terminus [10]. It has been well established that L-HDAg is essential for virion production [11]. Reports suggest that HDAg might contribute to HDV pathogenesis by regulating host signaling pathways [12,13].

HDV infection has two infection types: coinfection and superinfection [14]. Coinfection refers to individuals being infected with both HDV and HBV simultaneously, which may lead to severe acute hepatitis and liver failure. Superinfection, on the other hand, refers to individuals with pre-existing chronic HBV infection becoming infected with HDV again, accelerating the progression of cirrhosis, and in some studies, superinfected individuals with HDV have been shown to have a threefold increase in the risk of hepatocellular carcinoma compared to those with chronic HBV infection [15]. Eight genotypes have been identified based on sequence variations in HDV, with differences of up to 35% between different genotypes [16]. These genotypes exhibit regional variations in distribution, with the latest data indicating that genotype 1 predominates globally, while other genotypes are more localized and associated with different geographic regions. HDV-2 and HDV-4 are primarily found in Asia, HDV-3 is common in northern South America, and HDV-5 through HDV-8 are typically seen in people of African descent [17]. Different genotypes of HDV are believed to have different pathogenic potentials, with genotype 3 being considered the most pathogenic and being associated with outbreaks of hepatitis in South America [18]. In Taiwan, patients infected with genotype 1 tend to have more adverse outcomes compared to those infected with genotype 2 [19]. Recent studies estimate that approximately 12 million people worldwide have been infected with HDV [20], but the prevalence of HDV does not necessarily correlate directly with that of HBV [21,22]. The prevalence of HDV is a dynamic process that may be influenced by various factors such as HBV vaccination programs, public health infrastructure, and improvements in sanitation and hygiene [20,21,22].

An HDV database, vHDvDB 1.0, established in 2015, has been used to identify a naturally occurring HDV-4/4M intersubtype recombinant from 237 full-length sequences retrieved from GenBank, using phylogenetic and recombination analyses [23]. Subsequently, Miao et al. (2019) used a similar strategy to collect 345 full-length HDV sequences, identifying 31 new HDV RNA recombinants [24]. Recently, a comprehensive HDV database containing 512 complete HDV genome sequences established and published [25]. This database was utilized to predict the secondary structure of HDV RNA [26]. Furthermore, the increasing isolation of HDV sequences in recent years underscores the need for an accurate and up-to-date HDV sequence database for research purposes. To address this, we have expanded our original HDV full-length sequence database, vHDvDB 1.0 [23], to vHDvDB 2.0. Despite the abundance of data available on GenBank, the lack of systematic organization and classification leads to discrepancies in sequence annotations. Our research revealed that HDV researchers worldwide do not follow a consistent protocol when submitting HDV sequences. The most common issue identified was whether the amber/W editing site was reported as edited or unedited, which impacts whether the translated delta antigen is in its small or large form. Some HDV sequences submitted to GenBank appeared as antigenomes rather than the more common genomic sequences. Additionally, in some HDV sequences, the +1 position of nucleotide numbering differed from the commonly used position, as suggested by Wang et al., 1986 [4]. In this report, we collected and examined updated full-length HDV sequences to ensure they adhered to consistent specifications for systematic analysis and comparison. We manually completed the missing columns by gathering information commonly used in HDV studies and established a workflow to extract the amino acid sequences of both forms of the HDAg from the GenBank database. Additionally, we conducted secondary structure predictions for all the HDV sequences included in our updated database, vHDvDB 2.0. This database can generate sequence logos for selected HDV sequences, aiding researchers in studying the consistency and uniqueness among sequences. We also manually supplemented the geographical information for each sequence, labeling them according to their respective countries to create HDV sequence distribution maps for epidemiological research. Besides providing consistent basic information regarding HDV virology and epidemiology, this report also introduces an HDV RNA sequence checker to verify whether the nucleotide numbering of initially submitted sequences aligns with the standard numbering.

## 2. Materials and Methods

### 2.1. Database and Web Interface Implementation

We utilized the PHP Laravel framework to develop a semi-automated program for downloading and updating gbk files from GenBank using accession IDs. We employed open-source tools for sequence data processing, HDAg electric point calculation, HDV RNA secondary structure prediction, generation of HDAg sequence logos, and geographical distribution visualization. JavaScript libraries were used for frontend data visualization. Through the NCBI API, we downloaded complete HDV genome sequences from GenBank and manually filtered and extracted accession IDs, genotypes, and geographical information. After preprocessing the data, we analyzed and resolved issues such as incomplete genome data, naming confusion, and missing data for the large antigen and the small antigen. Through manual analysis, classification, transformation, and imputation, we successfully extracted relevant data for the large antigen and compared it with GenBank data to fill in the missing parts. HDV sequences were subjected to secondary structure prediction using the RNAfold [27] tool. The prediction results were recorded in dot–bracket format and then converted into SVG format images using RNAplot [28]. We manually extracted geographical information related to HDV sequences from literature reviews, using countries as the basis for classification. Our online system organized this information, grouping it by country, and displayed the distribution of selected HDV sequences on a world map. The map functionality utilized Leaflet to present the geographical distribution of HDV sequences, with larger circular markers indicating a higher number of different HDV sequences discovered in that region. This allows users to clearly and visualize the distribution of HDV sequences globally and quickly understand their distribution patterns.

### 2.2. HDV Sequence Data Collection and Correction

To calculate the amino acid sequence of the large antigen, we first extracted HDV full-length genomes from GenBank. We implemented a strategy to locate the start codon, RNA editing sites, and stop codon within the genomic sequences (Figure 1) by applying the following ten processing steps:I.All AUGs and their positions in the antigenome are designated Point A.II.All CC.UAG/CC.UGG/CC.URG sequences and their positions in the antigenome are designated Point B.III.All stop codons [UAA/UAG/UGA] and their positions in the antigenome are designated Point C.IV.The number of Point A positions must not exceed the number of Point B positions.V.The number of Point B positions must not exceed the number of Point C positions.VI.The lengths of regions A-B and B-C must be divisible by 3 (in-frame only).VII.The number of Point A positions must not exceed 90.VIII.The number of Point B positions must be greater than 500.IX.CC.UAG and CC.URG at Point B must be modified to CC.UGG.X.The length of region A-C must be between 600 and 660.

After the target nucleotides successfully extracted from the sequence are, potential proteins are calculated using the ORF function provided by Biopython, which is a Python programming library specified developed for biological purposes, and then unsuitable results are filtered based on three rules:I.The first amino acid of the generated protein sequence must be methionine.II.The fourth amino acid from the end of the protein sequence must be cysteine.III.The length of the protein sequence must be greater than 200.

In role II, this requirement is crucial for the functionality of the HDV ribozyme, as it contributes significantly to the enzyme’s structural integrity and catalytic activity. Previous studies have demonstrated this importance [29].

## 3. Results

### 3.1. Sequence Data

In previous studies, we identified partial sequences associated with HDV through RNA recombination [23]. Following this, we released vHDvDB 1.0, which initially included a partial set of HDV sequences for RNA recombination studies. In this study, the inclusion of HDV sequences was expanded to 732, through a systematic literature review, encompassing 498 HDV-1, 31 HDV-2, 59 HDV-3, 42 HDV-4, 22 HDV-5, 17 HDV-6, and 47 HDV-7, along with 9 HDV-8 and 1 recombined HDV sequence, totaling 726. The detailed statistics are provided in Table 1. Subsequently, we conducted data preprocessing by retrieving relevant files from GenBank using the NCBI API.

### 3.2. vHDvDB 2.0 Web Interface

vHDvDB 2.0 is a file system-based database. We extracted sequence files from the GenBank database based on manually filtered HDV sequences and processed them through a backend pipeline for collection, integration, supplementation, and preprocessing. The database architecture is illustrated in Figure 2. Our website provides a user-friendly interface enabling researchers to easily retrieve and compare HDV virus genomes. We updated and expanded the previous version of vHDvDB database in order to offer comprehensive sequences and more analytic functional modules, including new features to fulfill researchers’ needs for in-depth HDV genomics and proteomics studies. 

Additionally, we developed an RNAchecker for circular genomic viruses in sequence correction, and the corrected sequences are converted into gbk and fna file formats for downloading. We obtained relevant HDAg sequence information from the GenBank database, including their types, Protein IDs. Additionally, we added three new fields to the vHDvDB interface to display this information. Furthermore, we introduced four new tools at the top of the webpage: “Secondary Structure of HDV Genome”, “Gene Logo of Delta antigen”, and “World Map”. Users can select the HDV sequences and execute the systematic comparison.

### 3.3. Secondary Structure

We utilized RNAfold [27] to predict the secondary structures of 746 HDV genomes and their antigenome sequences. RNAfold, a part of the ViennaRNA package, employs an algorithm based on dynamic programming to compute the minimum free energy (MFE) structures. Subsequently, the predicted results were processed through RNAplot [28] to generate SVG-formatted secondary structure images, which were then converted to PNG format for display on the web. To smoothly display the oversized images of the RNA secondary structure, we used the gdal2tiles tool in Python to convert the images into a tiled map service format and uploaded them to the web server. Additionally, we employed the Leaflet tool to visually display these large secondary structure images on the webpage, providing users with a good user experience. Figure 3 shows the secondary structure prediction of the antigenome for AB037947. We have highlighted the nucleotide range of the large antigen in red to increase the readability.

### 3.4. Sequence Logo

A sequence logo is a visualization tool used to represent patterns in multiple sequence alignments. The height of each nucleotide or amino acid residue at each position represents the relative abundance of a certain nucleotide or amino acid at that position. Through sequence logos (Figure 4), we can more accurately describe sequence similarity and understand the structure or function of biological sequences. We have developed an online system that generates sequence logos based on the user’s selection of large or small antigen. Before generating the sequence logo for HDV sequences, the system first performs multiple-sequence alignment using ClustalW and then uses WebLogo (Version 3.7.12) to quickly generate the sequence logo, which it can convert to SVG format for better presentation on web interfaces.

We provide users with the ability to generate sequence logos for different HDV sequences selected in the database. When a user selects an HDV sequence, the system first performs multiple-sequence alignment using ClustalW and then generates the sequence logo in real-time using WebLogo, presenting it on the webpage in SVG format. By observing the sequence logo, we can understand the relative frequency of various nucleotides at specific positions in the HDV sequence, which is crucial for studying whether sequences among different genotypes are unique. Figure 4 demonstrates the sequence logo output.

### 3.5. World Map

The study completed geographical information and genotyping for 600 out of 746 HDV sequences through manual literature review and data organization from Genebank. Users can select different HDV sequences from the database and view their geographical locations on a world map to further investigate the transmission routes and epidemiology of HDV. The map categorizes the countries’ information based on HDV sequence, with each circle representing the number of HDV sequences found within a country. The size of the circle indicates the quantity of sequences, with larger circles representing more sequences. Users can click on the circles to display information such as sequence accession ID and genotype for the selected country. Figure 5 shows the geographical distribution and basic data of the selected HDV sequences.

### 3.6. RNA Sequence Checker

The RNA sequence checker is a tool designed specifically to check whether the initial positions of circular RNA sequences meet the standard, addressing the issue of sequence irregularities encountered by the existing vHDvDB. The workflow is illustrated in Figure 6, and the tool interface is shown in Figure 7. Through this checker, users can quickly and accurately compare whether different circular RNA sequences are in the same position, thereby determining their similarity. We present this checker in the form of a webpage, providing users with a simple and clear online interface and adjusting the presentation and professional terminology of the entire webpage to better suit the usage habits and academic needs of researchers, helping them explore the similarity between RNA sequences and make sequence comparisons more convenient. Additionally, in this web version of the checker, we have integrated several new features, such as one-click adjustment of uploaded sequences to +1 alignment and providing users with the ability to download files of different types, to meet the needs of researchers for a more in-depth study of RNA sequences. Furthermore, we are committed to continuous updates and improvements to ensure that users can enjoy the highest quality user experience and make a greater impact in the field of RNA sequence research.

The RNA sequence checker is a tool that utilizes a reference sequence M21012 [30] as the starting point for comparing RNA sequences uploaded by users. Once the user uploads a sequence, the checker performs a series of checks to ensure that the sequence conforms to the common length standards for HDV. Subsequently, it compares the uploaded sequence with M21012, allowing users to manually adjust the starting point if needed. Finally, users have the option to download the adjusted sequence files, including GBK and FASTA files, the CDS sequence of the sequence, and the protein sequence translated from the sequence. These options cater to the diverse needs of users, enabling them to conveniently download the required data and conduct subsequent research more effectively.

### 3.7. Database Updating and Maintenance

For continuous data updates and maintenance in the database, we use an automated backend program. When new viruses are added, the team only needs to manually enter the GenBank accession ID and geographical information. The backend program will automatically download the gene sequences and calculate data such as secondary structure and PI. The sequences can then be manually checked for accuracy using the workflow in Figure 6 before being imported into the database. Figure 8 explains the complete process of updates and review.

## 4. Discussion

A key focus of this paper is to conduct a comprehensive examination of the full-length HDV sequences uploaded to GenBank to ensure consistent representation. This includes checking and standardizing the +1 position of the HDV genomic sequences; standardizing the nucleotide numbering system for both the HDV genome and antigenome; and translating the nucleic acid sequences into amino acids for the HDAg. We also annotated the genotype and geographic location for each sequence.

vHDvDB 2.0 also includes RNA secondary structure analysis functionality. With the standardization of the nucleotide numbering system for the HDV genome and antigenome, it will facilitate the search for functional regions on the antigenome. We also annotated these functional regions in the HDV RNA structure. For example, the secondary structures near the RNA editing site and ribozymes in different sequences can be quickly searched on the website.

With the information and results obtained, we can use previously published methods to identify new naturally occurring HDV recombinants [9]. Moreover, we can conveniently obtain nucleotide logos and protein logos for each genotype. When the information from the eight genotypes is compared, it will aid in identifying molecular signatures specific to different genotypes. This, in turn, will help us identify new HDV recombinants through the identification of different molecular signatures.

In addition to creating a database and user-friendly interface, we have also developed a sequence correction workflow and automation processes. As more sequences are published in the future, we will continue to update the database content.

## 5. Conclusions

Our study effectively addressed issues of confusion in GenBank sequence data and unveiled several novel insights during the data preprocessing phase. We have eliminated the need for manual confirmation of each sequence by utilizing our developed RNA genome checker webpage. Furthermore, we have incorporated features such as semi-automatic correction of start site of HDV’s circular RNA sequences to the standard reference genome and the ability to download multiple file types. Additionally, in terms of frontend technology, we expanded vHDvDB by introducing several new sequence analysis tools. These tools include displaying predicted secondary structures, real-time generation of sequence logos, and a global distribution map of HDV sequences. In summary, our research not only successfully established a value-added database with practical utility but also developed a series of frontend technical tools. These advancements will facilitate deeper research within the HDV sequence domain for research teams.

## Figures and Tables

**Figure 1 viruses-16-01254-f001:**
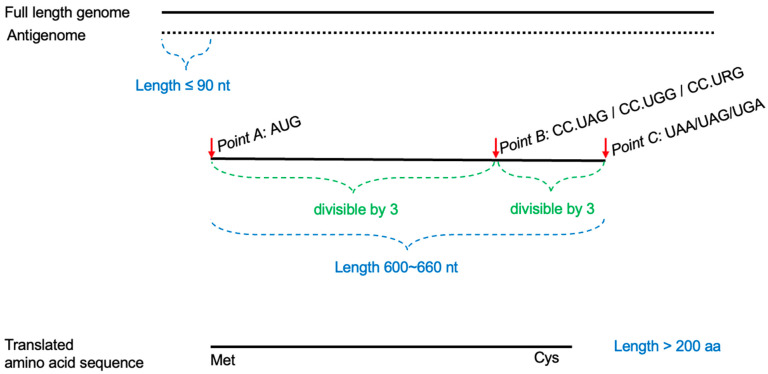
The sequence curation steps of vHDvDB 2.0. Based on the sequence processing rules mentioned earlier, we processed the full-length HDV genomic sequences downloaded from the database according to common sequence conventions for HDV viruses.

**Figure 2 viruses-16-01254-f002:**
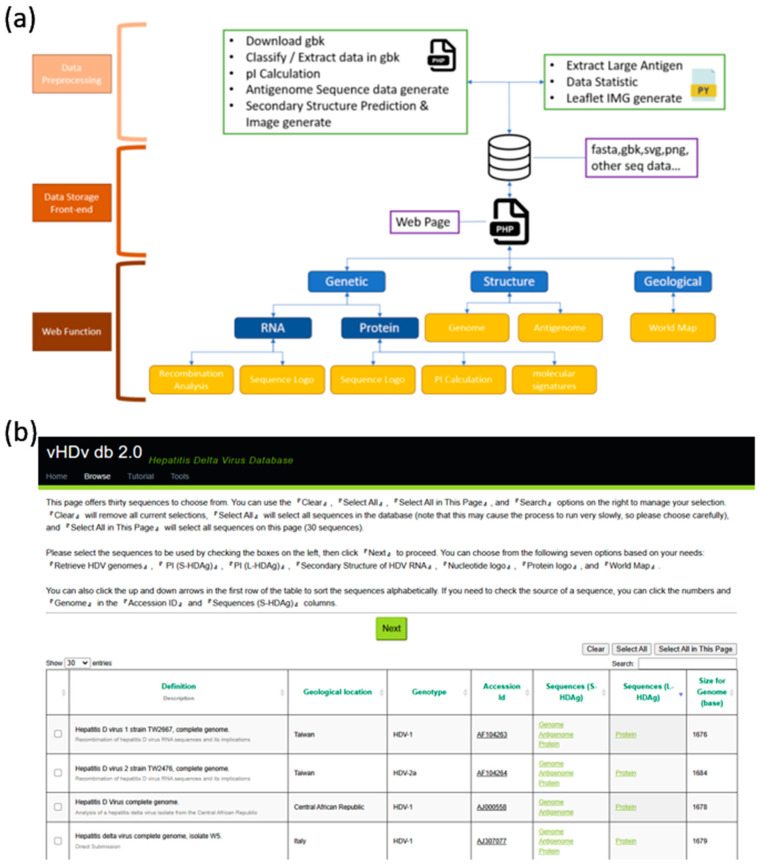
vHDvDB system architecture and web interface. (**a**) Relevant files are downloaded from GenBank’s API based on AccessionID using PHP programs, and then a series of PHP and Python programs preprocess the data, which are stored on the server in file format. (**b**) The web interface vHDvDB 2.0 primarily consists of dynamic searchable table as its main component. Additionally, we have integrated four new sequence analysis tools, including secondary structure prediction, sequence logo generation, and a geographical world map, allowing users to select relevant sequences for analysis as needed.

**Figure 3 viruses-16-01254-f003:**
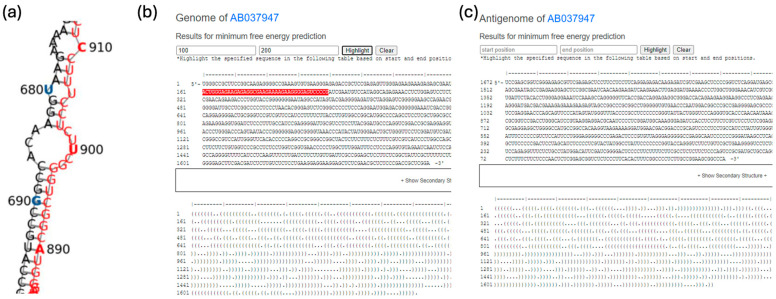
The predicted secondary structure of the antigenome for AB037947. (**a**) vHDvDB 2.0 has pre-computed the secondary structure of the sequence and generated the image. Additionally, every 10th nucleotide is highlighted in blue with position information marked alongside. The potential ranges of start codons and stop codons are highlighted in red. (**b**) vHDvDB 2.0 also presents the calculated secondary structure in dot–bracket notation on the webpage. Users can export the dot–bracket notation into other secondary structure visualization tools. Users can highlight selected sequence region by enter the start and stop position in the textbox. This feature can help to compare sequence regions among different viral genomes. (**c**) The antigenome view of queried sequences.

**Figure 4 viruses-16-01254-f004:**
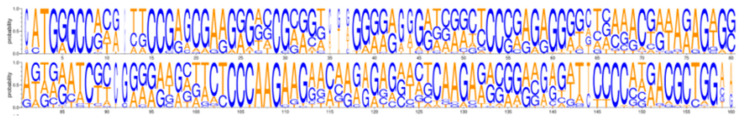
Sequence logo on vHDvDB 2.0. A sequence logo facilitates the analysis of sequence characteristics based on the user-selected sequence, according to research requirements.

**Figure 5 viruses-16-01254-f005:**
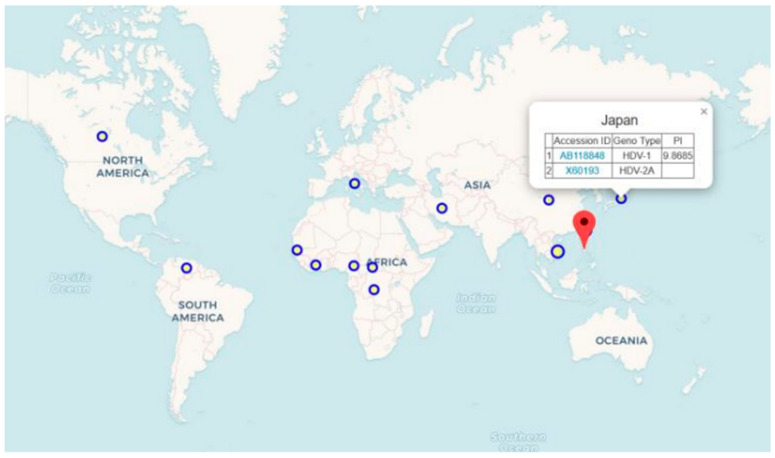
HDV world map viewer. The world map viewer displays the distribution of HDV sequences along with basic sequence information collected. Sequences are categorized by country using geographical information collected and sequences selected by users. By utilizing the interactive map, users can easily visualize the distribution of sequences and their related collections, which serves as a valuable reference for epidemiological research.

**Figure 6 viruses-16-01254-f006:**
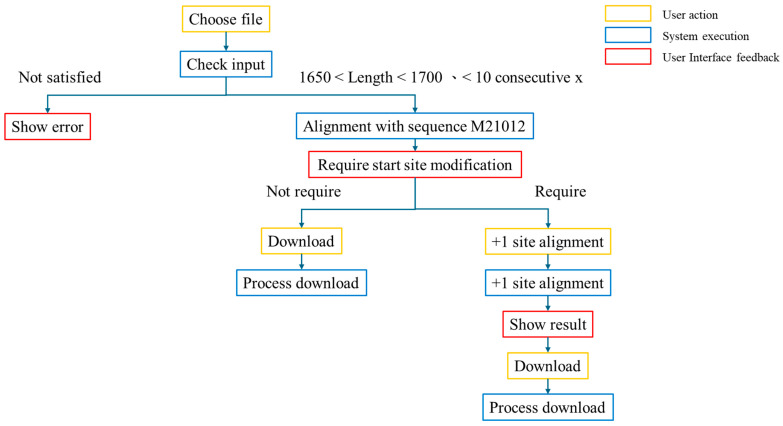
The workflow of the RNA sequence checker. The RNA sequence checker first checks the uploaded file to ensure it is in the gbk format. Then, the system verifies the length of the RNA sequence, requiring it to be greater than 1650 (all sequences in this database have lengths greater than 1650) to meet the evaluation criteria. Finally, we check the integrity of the uploaded sequence. If more than 10 consecutive Xs are found in the sequence, it is deemed incomplete. If the user’s sequence does not meet any of these criteria, the system prompts the user to upload another sequence that complies with the specifications. This process ensures that the uploaded sequences have the required file type, length, and integrity, thus ensuring the accuracy and reliability of subsequent analyses. After conducting the initial check on the RNA sequence, we align the user-uploaded sequence with the M21012 sequence. This alignment process compares the two sequences to determine their similarities and differences. The system utilizes the blastn algorithm for alignment and selects the best-performing alignment result between the two sequences for display to the user. The result is presented in a concise and clear chart, allowing users to intuitively understand the relative positions and similarity levels between the two sequences.

**Figure 7 viruses-16-01254-f007:**
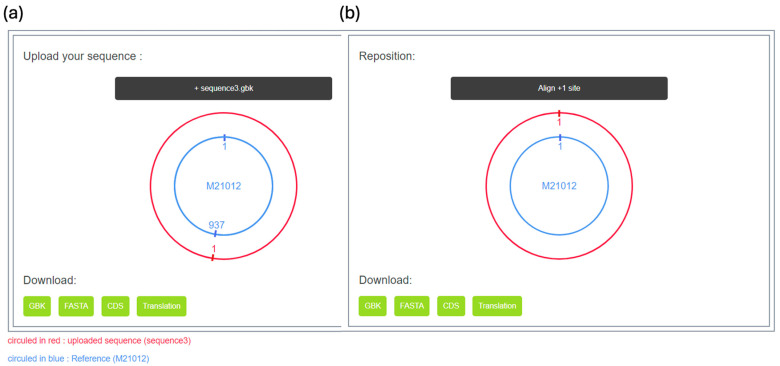
The web interface of the RNA checker. While displaying the alignment results (**a**), we provide two options for users to choose from. Users can adjust the +1 site (**b**), which means the system adjusts the user’s sequence based on the previous alignment results to maintain consistency with the M21012 sequence. This helps ensure the accuracy and reliability of the sequence and provides a reference point for easier subsequent analysis by researchers. Another option is to download the original sequence file, allowing users to download the file of their uploaded raw RNA sequence for further analysis or storage. Such options offer greater flexibility and convenience, enabling users to proceed with subsequent processing according to their needs.

**Figure 8 viruses-16-01254-f008:**
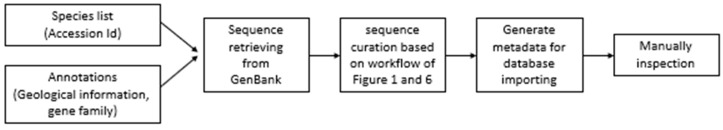
The database updating and maintaining workflow. Whenever a new virus strain is published, related IDs and annotation information are manually aggregated. Our automated program then performs post-processing. After two sequence curation processes and a manual check, the data are released to our database.

**Table 1 viruses-16-01254-t001:** The genotypes in vHDvDB 2.0. The data were extracted by the research team from existing literature and sequence files. The labels 2A and 2B, as well as 4A and 4B, represent subtypes of HDV-2 and HDV-4.

Genotype	Count
HDV-1	496
HDV-2A	28
HDV-2B	3
HDV-3	59
HDV-4A	17
HDV-4B	25
HDV-5	22
HDV-6	17
HDV-7	45
HDV-8	9
Recombinant HDV genotype I and II	1
No data	6
Total	728

## Data Availability

The website for accessing the database can be accessed for free through the following URL: http://ccllab.cgu.edu.tw:58031/ (accessed on 27 July 2024).

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
