# Peer review of "vHDvDB 2.0: Database and Group Comparison Server for Hepatitis Delta Virus"

_viruses, 2024, doi:10.3390/v16081254_

Round 1

Reviewer 1 Report

Comments and Suggestions for Authors

I have read the “vHDvDB 2.0: Database and Group Comparison Server for Hepatitis Delta Virus” draft and provide my review.

HDV (hepatitis D virus) infection is a serious global health problem that requires increased attention and research. The introduction of a new HDV database, as described in the article, can greatly advance HDV research and help scientists. By providing a centralized repository of HDV-related data, the database may allow scientists to conduct larger-scale studies, identify trends and patterns, and develop more effective prevention and treatment strategies.

However, there are some objections that should be pointed out. In general, the authors downloaded genomes from NCBI and added a few features to the database site. I don't see any scientific novelty in this development. At least for now, the site looks very crude and awkward, and hardly scientists will use it in their work with HDV.

Regarding text:

Line 14. “Hepatitis D virus (HDV) is a unique pathogen with significant global health implications, especially among individuals coinfected with Hepatitis B virus (HBV).”

Since hepatitis D infection cannot occur in the absence of hepatitis B virus, word "especially" looks unsuitable.

Line 131. “The fourth amino acid from the end of the protein sequence must be Cysteine.”

I think authors should provide some explanation for this rule because it is not clear for reader without solid HDV background; for example, it may be reference doi: 10.1128/JVI.79.2.1142-1153.2005.

Line 142. “Our previous released vHDvDB 1.0, initially included a partial set of HDV sequences for RNA recombination studies [9]”.

The reader may assume that the reference contains information about vHDvDB 1.0. However, this is not the case, and the help only contains information about HDV recombinants. In general, I have not found any mention of the vHDvDB database, either 1.0 or 2.0, on the Internet.

Overall, I cannot recommend this article for acceptance by the journal in this form and suggested that the authors supplement it with database driven research to show its potential.

Author Response

Response:

Thank you for the reviewer's suggestions and comments.

Indeed, the website's data is sourced from the NCBI database. However, when we previously used the NCBI database to search for HDV virus sequences, we often found that there were many sequences without distinguishing between full genome or partial genome categories. Although the sequences can be downloaded and then distinguished and compared, we believe that many biologists without an information background expect convenience in sequence queries and familiarity with related analysis and comparison tools.

If there are convenient analysis and comparison tools, the time required for sequence comparison can be greatly reduced. Additionally, we also found that HDV sequences in the PubMed database uploaded by different teams may have some differences in initial settings and conditions. For example, the initial melting points of the circular structure of HDV may differ among different teams, which requires calibration during subsequent sequence comparisons. However, such calibration methods may take more time for personnel who are not familiar with computer programming languages.

Therefore, we manually curated all the sequences, with specific procedures explained in Figure 1. We provide two types of proteins, Large and Small antigens, to facilitate researchers in directly comparing Large/Small antigens. Additionally, we have pre-prepared sequences for Genome and Antigenome, allowing users to directly view the secondary structure. Figure 3 of the manuscript is a convenient and fast short sequence search tool that can quickly highlight the input area range with colors.

Figure 1 (of the manuscript) illustrates the sequence curation workflow. By curating sequences in our database, the data can be directly utilized for analysis with greater efficiency.

Figure 3 (of the manuscript) provides short sequence search.

In terms of website functionality and smoothness, we have significantly enhanced the overall workflow in the revised version by reorganizing the layout and using visual color blocks to guide users in utilizing our database and the analysis comparison services provided on the website. For each functional block, we have provided guidance and clear icons. On the Browse page, users can select the sequences they want to compare, then click "next," which will bring up an analysis block with various comparison functions. Users can then click the buttons, and each function will be presented in a new window. This reduces user errors, while the species they selected will be retained on the first page. Additionally, we have added the function of uploading sequences. For some laboratories that can sequence new species themselves, they can upload these sequences to our website for comparison with existing sequences. Furthermore, we have redesigned the Tutorial page, explaining each service on the database with simple and easy-to-understand examples.

Regarding text:

Line 14. “Hepatitis D virus (HDV) is a unique pathogen with significant global health implications, especially among individuals coinfected with Hepatitis B virus (HBV).”

Since hepatitis D infection cannot occur in the absence of hepatitis B virus, word "especially" looks unsuitable.

Response:

Thank you for your corrections. We have made the changes in the article.

The revised sentence is on Page 1, line 14: The Hepatitis D virus (HDV) is a unique pathogen with significant global health implications, af-fecting individuals who are coinfected with the Hepatitis B virus (HBV).

Line 131. “The fourth amino acid from the end of the protein sequence must be Cysteine.”

I think authors should provide some explanation for this rule because it is not clear for reader without solid HDV background; for example, it may be reference doi: 10.1128/JVI.79.2.1142-1153.2005.

Response: Thank you for your suggestion. Indeed, this helps readers understand the reasoning behind the rule more quickly. We have already included this reference in the manuscript.

The revised sentence is on Page 3, line 135-137: In role II, this requirement is crucial for the functionality of the HDV ribozyme, as it significantly contributes to the enzyme's structural integrity and catalytic activity. Pre-vious studies have demonstrated this importance. [12].

Line 142. “Our previous released vHDvDB 1.0, initially included a partial set of HDV sequences for RNA recombination studies [9]”. The reader may assume that the reference contains information about vHDvDB 1.0. However, this is not the case, and the help only contains information about HDV recombinants. In general, I have not found any mention of the vHDvDB database, either 1.0 or 2.0, on the Internet. Overall, I cannot recommend this article for acceptance by the journal in this form and suggested that the authors supplement it with database driven research to show its potential.

Response: In our previous recombination research, we did indeed establish the first version of the database, which primarily provided a collection of sequences and download options, with the main analysis tools focusing only on recombination and phylogenetic tree construction for partial genome sequences. The newly published database builds on the previous version, enhancing manual sequence correction and continuous updates; it also offers a greater variety of sequence analysis tools.

The revised sentence is on Page 4 line 146-150: In previous studies, we identified partial sequences associated with HDV through RNA recombination [9]. Following this, we released vHDvDB 1.0, which initially included a partial set of HDV sequences for RNA recombination studies. In this study, the inclusion of HDV sequences was expanded to 732, through a systematic literature review, encompassing 498 HDV-1, 31 HDV-2, 59 HDV-3, 42 HDV-4, 22 HDV-5, 17 HDV-6, 47 HDV-7, along with 9 HDV-8 and 1 recombined HDV sequence, totaling 726.

Reviewer 2 Report

Comments and Suggestions for Authors

In their article, the authors propose an interesting research tool with the development of a HDV sequence database, completed with several data processing that facilitate HDV genome exploration. They very well described their sequence selection method from existing databases in network. Even if the article is clear and detailed, some information need to be clarify and checked, especially for selected sequences.

Major comments

1-      All the selected HDV sequences present in the URL cited in the article, need to be checked with great care because several errors are present. For example:

-          For 2 strains of HDV-2b genotypes, sequence is present in 3 times with 3 different accession numbers (Pt26 : JA417606-AX741209-AJ309880) and (Pt62 : JA417607-AX741210-AJ309879). Moreover JA417606 and JA417607 are not HDV-2a but HDV2-b genotypes.

-          For 1 strain of HDV-8 genotype, sequence is present in 3 times with 3 different accession numbers (dFr644 : AJ584849-AX741169-JA417566). Moreover JA417566 is not HDV-7 but HDV-8 genotype

-          For 1 strain of HDV-7 genotype, sequence is present in 3 times with 3 different accession numbers (dFr45 : AJ584844-AX741144-JA417541)

-          For 1 strain of HDV-6 genotype, sequence is present in 3 times with 3 different accession numbers (dFr48 : AJ584847-AX741164-JA417561)

-          For 3 strains of HDV-5 genotype, sequence is present in 3 times with 3 different accession numbers (dFr910 : AJ584848-AX741159-JA417556) and (dFr73 : JA417551-AX741154-AJ584846) and (dFr47 : JA417546-AX741149-AJ584845)

-          It misses one important HDV-3 sequence from Bolivia (AN: LT604954)

2-      It could be important to add the date of sample (strain) in this HDV database especially for researchers that want to perform molecular dating analyses. Does it be possible to add this information?

3-      The authors indicate sub-genotypes for example for HDV-2 strains. For HDV-1, the most representative genotype, they did not indicate sub-genotypes in their database (see table 1) whereas several ones were described for example in the following article (Hepatology. 2017 Dec;66(6):1826-1841. doi: 10.1002/hep.29574. Epub 2017 Oct 30. PMID: 28992360). In the same way, page 4 line 151, they indicate, “HDV-2 and -4 sub-genotypes have not been classified into sub-genotypes” whereas they were already defined in the above article, just as genotypes -5,-6,-7 and -8. Would it be possible to clarify sub-genotype classification for all genotypes in table 1, in accordance with existing classification?

Minor comments

1-      Page 1 line 37: The International Comity of Taxonomy of the Virus (ICTV) recently classified HDV into the Kolmioviridae family, thus it must be indicated in the sentence.

Author Response

Response:

Thank you for the reviewer's suggestions and comments.

Regarding the website's functionality and smoothness, we have made significant enhancements to the overall workflow in the revised version. By reorganizing the layout and incorporating visual color blocks, we guide users effectively in utilizing our database and the analysis comparison services provided on the website. For each functional block, we have included clear guidance and icons.

On the Browse page, users can select the sequences they wish to compare and then click "next." This action brings up an analysis block with various comparison functions. Users can click the corresponding buttons to access each function in a new window. This approach minimizes user errors, and the species selected are retained on the first page.

Additionally, we have introduced the capability to upload sequences. Laboratories that sequence new species independently can upload these sequences to our website for comparison with existing sequences. We have also redesigned the Tutorial page, providing simple and easy-to-understand examples explaining each service on the database.

1-      All the selected HDV sequences present in the URL cited in the article, need to be checked with great care because several errors are present. For example:

-          For 2 strains of HDV-2b genotypes, sequence is present in 3 times with 3 different accession numbers (Pt26 : JA417606-AX741209-AJ309880) and (Pt62 : JA417607-AX741210-AJ309879). Moreover JA417606 and JA417607 are not HDV-2a but HDV2-b genotypes.

-          For 1 strain of HDV-8 genotype, sequence is present in 3 times with 3 different accession numbers (dFr644 : AJ584849-AX741169-JA417566). Moreover JA417566 is not HDV-7 but HDV-8 genotype

-          For 1 strain of HDV-7 genotype, sequence is present in 3 times with 3 different accession numbers (dFr45 : AJ584844-AX741144-JA417541) 

-          For 1 strain of HDV-6 genotype, sequence is present in 3 times with 3 different accession numbers (dFr48 : AJ584847-AX741164-JA417561) 

-          For 3 strains of HDV-5 genotype, sequence is present in 3 times with 3 different accession numbers (dFr910 : AJ584848-AX741159-JA417556) and (dFr73 : JA417551-AX741154-AJ584846) and (dFr47 : JA417546-AX741149-AJ584845) 

-          It misses one important HDV-3 sequence from Bolivia (AN: LT604954)

Response:

Thank you for your careful corrections and valuable suggestions. We have updated our database to improve the accuracy of the sequences.

Pt26: Keep AJ309879, remove JA417606-AX741209; and Pt62: Keep AJ309880, remove JA417607-AX741210.

dFr644: Keep AJ584849, delete AX741169-JA417566 and change JA417566 to HDV-8 genotype.

dFr45: Keep AJ584844, delete AX741144-JA417541.

dFr48: Keep AJ584847, delete AX741164-JA417561.

dFr910: Keep AJ584848, delete AX741159-JA417556.

dFr73: Keep AJ584846, delete JA417551-AX741154.

dFr47: Keep AJ584845, delete JA417546-AX741149.

Correct LT604954 region to Tunisia.

2-      It could be important to add the date of sample (strain) in this HDV database especially for researchers that want to perform molecular dating analyses. Does it be possible to add this information?

Response:

Thank you for your suggestion. We have indeed considered adding a time field, as it would be very helpful for searching related chronological progressions, such as incorporating time information into the world map mode. However, the time information recorded in the database or that we can extract is not yet complete. Many of the sequence records' timestamps reflect the article publication date rather than the date the virus strain was discovered. We believe that adding time information to the database at this stage might potentially mislead readers.

3-      The authors indicate sub-genotypes for example for HDV-2 strains. For HDV-1, the most representative genotype, they did not indicate sub-genotypes in their database (see table 1) whereas several ones were described for example in the following article (Hepatology. 2017 Dec;66(6):1826-1841. doi: 10.1002/hep.29574. Epub 2017 Oct 30. PMID: 28992360). In the same way, page 4 line 151, they indicate, “HDV-2 and -4 sub-genotypes have not been classified into sub-genotypes” whereas they were already defined in the above article, just as genotypes -5,-6,-7 and -8. Would it be possible to clarify sub-genotype classification for all genotypes in table 1, in accordance with existing classification?

Response:

Thank you for your suggestion. We have made corrections on the website and updated the main text of the article (lines 149, 150, 151) and Table 1.

The revised sentence is on Page 4, line 157: Following this, we released vHDvDB 1.0, which initially included a partial set of HDV sequences for RNA recombination studies. In this study, the inclusion of HDV sequences was expanded to 732, through a systematic literature review, encompassing 498 HDV-1, 31 HDV-2, 59 HDV-3, 42 HDV-4, 22 HDV-5, 17 HDV-6, 47 HDV-7, along with 9 HDV-8 and 1 recombined HDV sequence, totaling 726. The detailed statistics are provided in Table 1.

Minor comments

1-      Page 1 line 37: The International Comity of Taxonomy of the Virus (ICTV) recently classified HDV into the Kolmioviridae family, thus it must be indicated in the sentence.

Response:

Thank you for your corrections. We have made the amendments to the text.

The revised sentence is on Page 1, line 37: HDV belongs to the Kolmioviridae virus family, which was recently classified by the International Committee on Taxonomy of Viruses (ICTV).
